# Nutritive Value and Bioactivities of a Halophyte Edible Plant: *Crithmum maritimum* L. (Sea Fennel)

**DOI:** 10.3390/plants13030427

**Published:** 2024-01-31

**Authors:** Iris Correia, Madalena Antunes, Carla Tecelão, Marta Neves, Cristiana L. Pires, Pedro F. Cruz, Maria Rodrigues, Claúdia C. Peralta, Cidália D. Pereira, Fernando Reboredo, Maria João Moreno, Rui M. M. Brito, Vânia S. Ribeiro, Daniela C. Vaz, Maria Jorge Campos

**Affiliations:** 1Marine and Environmental Sciences Centre, Polytechnic of Leiria, 2520-630 Peniche, Portugalmarta.neves@ipleiria.pt (M.N.); 2School of Tourism and Marine Technology, Polytechnic of Leiria, 2520-630 Peniche, Portugal; 3Coimbra Chemistry Centre (CQC), Institute of Molecular Sciences, Chemistry Department, University of Coimbra, 3004-535 Coimbra, Portugalpjfc7@ci.uc.pt (P.F.C.); mmoreno@ci.uc.pt (M.J.M.);; 4Laboratory of Separation and Reaction Engineering–Laboratory of Catalysis and Materials (LSRE-LCM), ESTG-IPLeiria, 2411-901 Leiria, Portugal; maria.l.rodrigues@ipleiria.pt (M.R.); vania.ribeiro@ipleiria.pt (V.S.R.); 5ALiCE–Associate Laboratory in Chemical Engineering, University of Porto, 4200-465 Porto, Portugal; 6School of Health Sciences, Polytechnic of Leiria, 2411-901 Leiria, Portugal; cidalia.pereira@ipleiria.pt; 7Centre for Innovative Care and Health Technology, Polytechnic of Leiria, 2411-901 Leiria, Portugal; 8GeoBioTec, FCT, Universidade NOVA de Lisboa, 2829-516 Caparica, Portugal; fhr@fct.unl.pt

**Keywords:** *Crithmum maritimum*, sea fennel, halophyte, centesimal composition, antioxidant capacity, polyphenolic content, antimicrobial properties, mineral profile, Caco-2 model cells, prebiotic effect

## Abstract

*Crithmum maritimum* L. (sea fennel), an edible xerophyte of coastal habitats, is considered an emerging cash crop for biosaline agriculture due to its salt-tolerance ability and potential applications in the agri-food sector. Here, the nutritional value and bioactive properties of sea fennel are described. Sea fennel leaves, flowers, and schizocarps are composed of carbohydrates (>65%) followed by ash, proteins, and lipids. Sea fennel’s salty, succulent leaves are a source of omega-6 and omega-3 polyunsaturated fatty acids, especially linoleic acid. Extracts obtained from flowers and fruits/schizocarps are rich in antioxidants and polyphenols and show antimicrobial activity against *Staphylococcus aureus*, *Staphylococcus epidermis*, *Candida albicans*, and *Candida parapsilosis*. Plant material is particularly rich in sodium (Na) but also in other nutritionally relevant minerals, such as calcium (Ca), chlorine (Cl), potassium (K), phosphorus (P), and sulfur (S), beyond presenting a potential prebiotic effect on *Lactobacillus bulgaricus* and being nontoxic to human intestinal epithelial Caco-2 model cells, up to 1.0% (*w*/*v*). Hence, the rational use of sea fennel can bring nutrients, aroma, and flavor to culinary dishes while balancing microbiomes and contributing to expanding the shelf life of food products.

## 1. Introduction

*Crithmum maritimum* L., also known as “sea fennel” and “rock samphire”, is an edible halophyte and xerophyte that grows wildly on the Mediterranean and Atlantic coasts of Europe [1,2]. This aromatic flowering plant of the Apiaceae family is considered a valuable emerging crop for sustainable agriculture in marginal environments, given its suitability to adapt to different climatic conditions and to biotic and abiotic stress [1,3]. Considering the increasing interest in diet diversification and demand for vegan, vegetarian, and ecologically labeled food, cultivation of halophytes and xerophytes offers a viable alternative against freshwater shortage and soil salinization and desertification [2,4,5,6]. Besides its interest as a salt-tolerant species, *C. maritimum* is a source of natural bioactive compounds that are economically relevant for the industries of food, pharmaceuticals, and cosmetics [1,7,8,9].

*C. maritimum* has been traditionally used by local communities in salads, soups, and sauces, as well as to prepare medicinal decoctions and infusions [10]. *C. maritimum* essential oils obtained from different parts of the plant (flowers, leaves, and stems) are known to be rich in antioxidants [11,12,13,14], antimicrobial [15,16,17,18], trypanocidal [19,20], acaricidal [21], insecticidal [22,23], and nematicidal [24] compounds, such as tannins, flavonoids, phenolic acids, and terpenes [25,26]. Some of these compounds are monoterpenes, such as *γ*-terpinene, limonene, and sabinene [13,14,17,19,24,26] phenylpropanoids, such as dillapiole [16,22,25], and the polyacetylene falcarindiol, a natural pesticide also found in carrots [27]. These antioxidant and antimicrobial properties of sea fennel’s essential oils have also been successfully transferred to sauces and non-dairy probiotics based on sea fennel fruits/schizocarps [28,29], and to enrich microemulsions to apply as natural preservatives, to increase the shelf stability of food products [30]. Thus, the aerial parts of sea fennel can be used in the agri-food sector as natural food additives or food preservatives that also bring health benefits, color, and aroma while valorizing biodiversity and preserving endemic species.

Nonetheless, even if the beneficial properties of the essential oils of sea fennel have been described and recognized, a complete description of the nutritional value of the various parts of this edible halophyte is still missing. Hence, here, the centesimal composition of different aerial parts of *C. maritimum* (leaves, flowers, and schizocarps), along with the description of its mineral composition, profile of fatty acids, chlorophylls, xanthophylls, and carotenoids is reported. In addition, the characterization of sea fennel extracts in terms of antimicrobial activity, antioxidant capacity, and phenolic content is also presented. Evaluation of the antimicrobial activity involving antibacterial testing against Gram-positive (*Staphylococcus aureus* and *Staphylococcus epidermidis*) and Gram-negative (*Escherichia coli* and *Klebsiella pneumoniae*) bacteria, as well as against pathogenic and non-pathogenic fungal strains (*Candida albicans*, *Candida parapsilosis*, *Komagataella phaffii/Pichia pastoris* and *Saccharomyces cerevisiae*) was also conducted. In addition, human intestinal epithelial Caco-2 model cells were used to test for cell viability upon direct application of plant material and the potential prebiotic effect of sea fennel leaves was also investigated on the growth of probiotic *Lactobacillus bulgaricus*.

## 2. Results and Discussion

### 2.1. Centesimal Composition and Mineral Content

Moisture contents of *C. maritimum* flowers, leaves, and fruits/schizocarps are shown in Table 1. Values ranged from 88.4 ± 0.7% in leaves to 85.0 ± 0.1% in flowers and are in line with those reported by Sánchez-Faure et al. [31] for tender stems of sea fennel collected at the north coast of Galicia, Spain.

Centesimal compositions are also shown in Table 1 and Appendix A, expressed as a percentage of dry weight (% dw). All plant parts of sea fennel are mainly composed of carbohydrates, followed by ash, protein, and crude fat, in agreement with data previously reported for tender stems of sea fennel and for leaves of other edible halophytes, such as common ice plant (*Mesembryanthemum crystallinum*) and seaside arrowgrass (*Triglochin maritima*) [31]. Sea fennel flowers are richer in proteins (8.3 ± 0.2% dw) and carbohydrates (71.7 ± 0.4% dw), while leaves are richer in ash (24.9 ± 0.8% dw) and schizocarps in lipids (6.5 ± 0.5% dw), given that they are mainly composed by the seed, and have almost no pulp.

A detailed analysis of the mineral composition of the aerial parts of *C. maritimum* is also shown in Table 1 and Appendix A. Contents in calcium (Ca), chloride (Cl), copper (Cu), iron (Fe), manganese (Mn), phosphorus (P), potassium (K), sodium (Na), sulfur (S), and zinc (Zn), were obtained per gram of dry weight of plant material (g dw). Sea fennel aerial parts are particularly rich in Na (36,500 ± 500.0 µg/g dw), Ca (1180 ± 10.0 µg/g dw), Cl (2980 ± 20.0 µg/g dw), K (1600 ± 20.0 µg/g dw), P (335 ± 49.0 µg/g dw), and S (983 ± 55.2 µg/g dw), as other edible succulent halophytes, such as iceplant (*Mesembryanthemum nodiflorum*) or sarcocornia (*Sarcocornia fruticose*), reflecting the high salinity environments inhabited by the plants, particularly the calcareous cliffs and rocky soils of the western coast of Europe/Portugal [32]. Although in high amounts, Na levels in sea fennel are lower than the Na contents found in *Salicornia ramosissima* (89,900 ± 500 µg/g dw) and *Sarcocornia perennis* (64,200 ± 900 µg/g dw) leaves [33].

### 2.2. Photosynthetic Pigments and Antioxidant Capacity

Amounts of chlorophylls, xanthophytes, and carotenes are also shown in Table 1 and Appendix A. Chlorophylls are present in higher amounts in all aerial parts, being particularly abundant in leaves when compared to xanthophylls and carotenes. The amount of pigments found in the flowers, leaves, and schizocarps of sea fennel is comparable to the amount found in the leaves of other edible halophytes, such as *Sarcocornia perennis* and *Salicornia ramosissima* [33], especially chlorophyll a, which is always more abundant in all plant parts.

The antioxidant capacity of sea fennel flowers, leaves, and schizocarps was determined by three redox methods (ABTS, DPPH, and FRAP assays) based on hydrogen atom transfer (HAT) and/or single electron transfer (SET) mechanisms. Extracts were prepared by sequential extraction in different solvents, namely, petroleum ether, followed by acetone, ethanol, 50% (*v*/*v*) ethanol, and water, as a way to obtain all water-soluble and insoluble antioxidants present in the plant material. Antioxidant capacities are shown in Table 2 and Figure 1, expressed in micrograms of Trolox equivalent antioxidant capacity per milligram of plant material (µg Trolox Eq/mg).

Comparing the results achieved with the several solvents, ethanol extracts stand out, leading to the strongest antioxidant activities, followed by those of 50% (*v*/*v*) ethanol, regardless of the part of the plant tested or method used, in contrast with the lowest values registered in the ether extracts. On the other hand, the acetonic and ethanolic extracts of flowers presented higher activity than the corresponding extracts of leaves and fruits. However, this pattern was not constant for the other solvents. In fact, extracts of leaves obtained with petroleum ether, 50% (*v*/*v*) ethanol, and water exhibited values slightly higher than those of flowers. Extracts of schizocarps always presented the lowest activities in the three assays performed. These data suggest that the compounds of *C. maritimum* with the strongest antioxidant activities are more abundant in flowers and have a moderate polarity, being preferentially extracted with ethanol.

Extracts obtained in petroleum ether showed similar antioxidant capacities by the three assays, with higher values detected by the DPPH scavenging method (especially in leaves), a radical known to detect hydrophobic compounds more efficiently, given its lipophilic character [34]. Plant material was then submitted to acetone, which was able to proceed with the extraction of more antioxidant compounds, especially from flowers, as detected by all methods, particularly by the SET-based nonradical method FRAP, which detects several types of polyphenolics [35]. Subsequent extraction with more hydrophilic solvents (ethanol, ethanol/water, water) led to the extraction of more antioxidants from the leaves and from the flowers, as detected by all assays (ABTS, DPPH, and FRAP), which show high correlation among them (Pearson’s coefficient ~0.95), as well as stated by the positive correlation between the amount of antioxidants extracted by ethanol and 50% ethanol (0.889) and by 50% ethanol/water (0.946). Thus, while leaves seem to be richer in more hydrophobic antioxidants (e.g., carotenoids and flavones), flowers are richer in more hydrophilic molecules (e.g., phenolic acids, flavanols, and anthocyanins).

### 2.3. Phenolic Content

Total phenolic contents (TPCs) were obtained by the Folin–Ciocalteu method. Results are expressed in Table 2 and Figure 1 in micrograms of gallic acid equivalents per milligram of plant material (µg GAE/mg). TPCs are within the range of previously reported data for other edible halophytes, such as salicornia and sarcocornia [33] or immortelle (*Helichrysum italicum*) [34]. TPCs were mainly extracted with more hydrophilic solvents (ethanol, ethanol/water, and water) and were more abundant in flowers (254 ± 15 µg GAE/mg, with ethanol), indicating that more hydrophilic compounds are present in all plant parts, especially in the flowers. Moreover, TPC values also positively correlate (Pearson’s coefficient) with the data obtained by the ABTS (0.957), DPPH (0.950), and FRAP (0.961) assays, reinforcing that most of the antioxidants present in the aerial parts of sea fennel possess phenolic moieties. Hence, 1D ^1^H and 2D [^1^H-^1^H]-COSY proton NMR experiments were also conducted to identify major groups of phenolic protons and their respective correlations. The downfield aromatic regions (δ 6.0 to 8.0 ppm) of the 1D ^1^H NMR spectra of sea fennel’s leaves, flowers, and schizocarps are shown in Appendix A. Phenolic compounds are typically present as doublet (d) signals assigned to ortho- and meta-protons of the hydroxybenzoic moiety (at chemical shift (δ), multiplicity, vicinal coupling constants ^3^J_HH_: 6.80 (d, J = 8.4 Hz), 6.95 (d, J = 2.0 Hz), and 7.00 (dd, J = 8.4 Hz, J = 2.0 Hz), and to protons of the acrylic group, in the case of hydroxycinnamic acids (7.50 (d, J = 15.9 Hz) and 6.25 (d, J = 15.9 Hz)) [36], such as caffeic acid, ferulic, and chlorogenic acids. Hence, as seen in Appendix A for the NMR spectra of sea fennel flowers, leaves, and schizocarps, as the sequential extraction proceeds to more hydrophilic solvents (ethanol > ethanol/water > water), more NMR signals assigned to protons of phenolic moieties can be identified and increased in intensity, corroborating the TPC results shown in Table 2 and Figure 1.

### 2.4. Fatty Acid Content

The fatty acid (FA) profile of leaves, flowers, and schizocarps of sea fennel (*C. maritimum*) is shown in Table 3. All plant parts are mainly composed of unsaturated fatty acids: leaves (66.55% of total FAs), flowers (74.52 of total FAs), and schizocarps (88.90% of total FAs). However, while leaves and flowers are mainly composed of polyunsaturated fatty acids (PUFAs) (higher than 50% of total FAs), schizocarps mostly contain monounsaturated (MUFAs) fatty acids (60.95 ± 0.76% of total FAs). Moreover, the amounts of the most abundant FA clearly differ between the different parts of the plant.

Hence, in leaves, linoleic (C18:2 n-6) and alpha-linolenic (ALA, C18:3 n-3) acids were the most abundant FAs detected, attaining values of 5.11 ± 0.33 µg/mg dw (29.47 ± 0.39% of total FA) and 3.90 ± 0.24 µg/mg dw (22.50 ± 0.36% of total FA), respectively, in line with other studies [31]. The amount of C18:2 n-6 found in flowers (11.77 ± 0.64 µg/mg dw, 29.47 ± 0.39% of total FA) was almost twice the values registered in leaves. In schizocarps, linoleic acid only represented 23.70 ± 0.50% of total FAs, but its absolute concentration (11.6 ± 0.33 µg/mg dw) is similar to those of flowers since schizocarps are richer in lipids than the other parts of *C. maritimum*.

In opposition, the values of the essential PUFA ALA registered in flowers (2.22 ± 0.12 µg/mg dw, 8.54 ± 0.04% of total FAs) and schizocarps (1.30 ± 0.41 µg/mg dw, 2.67 ± 0.10% of total FA) are clearly below of those of leaves. The eicosapentaenoic acid (EPA) was also detected, but in minor amounts, ranging from 0.74 ± 0.25 µg/mg dw in schizocarps to 0.091 ± 0.001 µg/mg dw in leaves.

Moreover, oleic acid (C18:1 n-9) was the most plentiful FA found in schizocarps (29.7 ± 2.0 µg/mg dw, 50.1 ± 1.0%), whereas erucic acid (C22:1 n-9) was the MUFA more abundant in leaves (1.139 ± 0.052 µg/mg dw, 6.62 ± 0.07%) and flowers (2.61 ± 0.16 µg/mg dw, 10.4 ± 0.17%), although in minor quantities than in schizocarps (3.4 ± 1.0 µg/mg dw, 7.05 ± 0.30% of total FA). Other MUFAs present in considerable amounts were nervonic (C24:1 n-9) and palmitoleic (C16:1 n-7) FA.

In turn, palmitic (C16:0) and stearic (C18:0) acids were the major saturated fatty acids (SFAs) found in *C. maritimum* with values of 4.359 ± 1.440 µg/mg dw and 0.200 ± 0.080 µg/mg dw in schizocarps, 4.570 ± 0.270 µg/mg dw and 0.918 ± 0.051 µg/mg dw in flowers and of 3.210 ± 0.130 µg/mg dw and 1.377 ± 0.062 µg/mg dw, in leaves, respectively.

Thus, all aerial parts of sea fennel have shown to be a source of nutritionally valuable FAs, with high PUFA/SFA and H/H ratios and low atherogenicity (AI) and thrombogenicity (TI) indexes comparable to other edible halophytes [31,34,37].

### 2.5. Antibacterial Activity

The antibacterial activity of the non-polar extracts of sea fennel leaves, flowers, and schizocarps was evaluated against two Gram-negative strains, *E. coli* ATCC 25922 and *K. pneumoniae* ATCC 27799, and two Gram-positive strains, *S. aureus* ATCC 29213 and *S. epidermidis* ATCC 12228. MIC (minimal inhibitory concentration) and MBC (minimal bactericidal concentration) of extracts are shown (in µg/mL) in Table 4. Extracts were more efficient in inhibiting the growth of the Gram-positive bacteria, *S. aureus* ATCC 2921 and *S. epidermidis* ATCC 12228, than that of the Gram-negative bacteria tested. Extracts obtained from flowers and schizocarps showed higher antibacterial activity, attaining MIC values as low as 75 µg/mL for *S. aureus* ATCC 2921 for extracts obtained in petroleum ether and acetone. In contrast, extracts obtained from the leaves showed an MIC against the tested bacteria generally higher than 600 µg/mL. The antimicrobial activity of petroleum ether and acetone extracts of flowers and schizocarps was similar. The bacterial growth of the two Gram-negative strains tested herein was only inhibited at high concentrations of extracts, in agreement with previously reported data for apolar extracts obtained from the seeds and aerial parts of sea fennel against *E. coli* [15,38,39]. For all extract and plant parts, the ratio MBC/MIC was close to one. When this ratio is less than two, the effect can be regarded as bactericidal.

In a study conducted by Souid et al. [40], significant antimicrobial activity of sea fennel leaves against *S. aureus* was observed in a hydroethanolic extract. In our case, however, the leaf extract of sea fennel demonstrated the lowest activity against this microorganism, whereas the flower and schizocarps extracts obtained with petroleum ether and acetone exhibited greater inhibitory effects on the growth of *S. aureus*. According to the same authors, this activity was attributed to phenolic compounds, such as the chlorogenic and neochlorogenic acids present in the plant. Previous studies have also shown that extracts of sea fennel seeds presented antibacterial activity against *Bacillus cereus* [38], while extracts of leaves presented antibacterial activity against *Pseudomonas* strains [15,39], possibly attributed to the presence of the compound falcarindiol [27], reinforcing the potential use of sea fennel as a natural preservative, as well as a medicinal plant.

### 2.6. Antifungal Activity

The antifungal activity of apolar extracts of sea fennel leaves, flowers, and schizocarps was tested against pathogenic and non-pathogenic fungal strains: *C. albicans* ATCC 10259, *C. parapsilosis* ATCC 22019, *S. cerevisiae* ATCC 18824, and a wild strain of *K. phaffii*. Minimal inhibitory concentrations (MICs) and minimum fungicidal concentrations (MFCs) of extracts are shown (in µg/mL) in Table 4. Extracts were efficient in inhibiting the growth of pathogenic fungi of the genus *Candida*, particularly *C. albicans* (MIC of 75.0 µg/mL and MFC of 150.0 µg/mL), in agreement with the results obtained by Houta et al. [41] that also reported antifungal activity by methanolic extracts of sea fennel against *C. albicans*.

As seen above for the antibacterial activity assays, extracts obtained from flowers and schizocarps showed higher antifungal capacity than extracts obtained from leaves. Interestingly, extracts obtained from sea fennel schizocarps in petroleum ether were more efficient than acetonic extracts in inhibiting the fungal growth of *C. albicans* and *C. parapsilosis*; while extracts obtained from flowers in acetone were more efficient than extracts in petroleum ether in inhibiting *C. albicans*. At the tested concentrations, the fungal growth of *S. cerevisiae* was not inhibited by extracts obtained from leaves or flowers (MICs > 600.0 µg/mL). However, the extracts obtained from schizocarps in petroleum ether and acetone showed some level of inhibition against *K. phaffii* and *S. cerevisiae*. These data also reinforce the potential use of sea fennel as a natural preservative and medicinal plant, as a way to extend the shelf life of food products and cosmetics, as performed by other plant extracts [42], given its bioactive properties and beneficial effects, as already stated by the positive use of sea fennel cell biomass in food products [43,44] and in skin repair [45]. Similarly, in line with the bactericidal effect of sea fennel extracts (MBC/MIC < 2), the MFC/MIC ratio also suggests that the primary effect against fungi is fungicidal.

### 2.7. Caco-2 Cells Viability

Caco-2 cells were used to examine possible toxic effects caused by the aerial parts of *C. maritimum*. Figure 2 shows the percentage of cell viability according to the MTT assay. After being submitted to various concentrations of plant material (as high as 5.0% (*w*/*v*)), Caco-2 cells still presented viability higher than 50% when compared to the control. Consumption of herbs and spices is an average of 0.4 g per person per meal (0.5 to 1.0% (*w*/*v*)) [46]. Thus, when included in a meal, the availability of the plant material for intestinal absorption is expected to be further decreased, thus leading to no significant toxicity to cells of the intestinal epithelium.

Previous studies performed with other human cell lines have also explored the medicinal value of sea fennel essential oils and leaf extracts. Ethyl acetate extracts of *C. maritimum* have been shown to inhibit the growth of hepatocellular carcinoma (HCC) cells by reducing intracellular lactate, inhibiting protein anabolism, and affecting lipid homeostasis (reduction of triglycerides, cholesterol, MUFAs, and PUFAs). These findings highlight the role of sea fennel as a promising tool for the prevention and treatment of hepatocellular carcinoma [47]. Additionally, sea fennel aqueous extracts were loaded into soybean phosphatidylcholine liposomes and delivered to Caco-2 cells. As a result, cells presented anti-inflammatory activity (by decreasing their pro-inflammatory response) via lowering levels of interlekin-10 (IL-10) and tumor necrosis factor alpha (TNF-α). Thus, sea fennel can be a valuable nutraceutical tool given its phytochemical profile and antioxidant, antimicrobial, and anti-inflammatory properties.

### 2.8. Prebiotic Effect on Lactobacilli

The potential prebiotic effect of sea fennel leaves was also investigated on the growth of probiotic *Lactobacillus bulgaricus*. Bacterial growth rates were different in the absence and presence of *C. maritimum* plant material at 0.1% (*w*/*v*). *Lactobacilli* growth was faster in the presence of sea fennel leaves, as shown by the bacterial growth curves after 12 h of incubation at 37 °C (Appendix A), indicating that the plant may hold prebiotic potential. Previous authors have also shown that *C. maritimum* may be a potential vehicle for probiotic (*Lactiplantibacillus plantarum* IMC 509) delivery to humans by making use of an artificially acidified, pasteurized sea fennel preserved in brine submitted to 44 days of refrigeration [29]. Thus, the sea-fennel-derived nutrients present in the leaves (with 66.5% carbohydrates) may support the survival of the probiotic strains, as also seen for other probiotic-enriched foods [48].

## 3. Materials and Methods

### 3.1. Plant Material and Analytical Reagents

*Crithmum maritimum* plant material was hand-collected during September 2021 at the marine cliffs of Cruz dos Remédios (Peniche, Portugal), 39°21′57″ N 9°24′15″ W, 21 m above sea level and taxonomically identified. Once harvested, plant materials were cleansed with ultrapure Milli-Q water (Millipore Co., Burlington, MA, USA), separated accordingly (aerial parts, leaves, flowers, schizocarps) frozen at −80 °C (Haier Medical, Qingdao, China, DW-86L728J), freeze-dried (Telstar, Terrassa, Barcelona, Lyoquest-85), ground to powder, and stored at −20 °C, until use.

All chemicals were of analytical grade or higher. ABTS ([2,2′-azinobis(3-ethylbenzothiazoline-6-sulfonic acid)]-diammonium salt), acetone, agar, cefotaxime, D-glucose, DMEM (Dulbecco’s modified Eagle’s medium high glucose), DPPH (2,2-diphenyl-1-picrylhydrazyl), ethanol, fluconazole, gallic acid, MTT (3-(4,5-dimethylthiazol-2-yl)-2,5-diphenyl-tetrazolium bromide), non-essential amino acids (10×), peptone, petroleum ether, fatty acid methyl esters standards (Supelco 37 Component FAME Mix, C17 FAME; PUFA 1 and PUFA 3), TPTZ (2,4,6-tris(2-pyridyl)-s-triazine), Trolox (6-hydroxy-2,5,7,8-tetramethylchroman-2-carboxylic acid), trypsin–EDTA solution (10×), and yeast extract were obtained from Sigma-Aldrich (Merck KGaA, Darmstadt, Germany). Folin–Ciocalteu reagent was purchased from Biochem Chemopharma (Nevers, France). Sodium carbonate was acquired from Scharlau (Scharlab, Barcelona, Spain). Human colon adenocarcinoma cell line Caco-2 was obtained from the European Collection of Cell Cultures (ECACC, U.K.). Bacterial and fungal strains used were *Escherichia coli* ATCC 25922, *Klebsiella pneumoniae* ATCC 27799, *Staphylococcus aureus* ATCC 29213, *Staphylococcus epidermis* ATCC 12228, *Candida albicans* ATCC 14053, *Candida parapsilosis* ATCC 22019, *Saccharomyces cerevisiae* ATCC 18824 and a wild strain of *Komagataella phaffii*. RPMI 1640 synthetic medium, potato dextrose agar, and fetal bovine serum (FBS) were acquired from Thermo Fisher (Thermo Fisher Scientific Inc., Waltham, MA, USA). Kjeldahl tablets (5 g without Se and Hg) were acquired from Merck (Germany). Isotope-labeled compounds were purchased from Cambridge Isotope Laboratories Inc. (Tewksbury, MA, USA). Sulfuric acid, 95–97%, and nitric acid, 65%, were purchased from Chem-lab (Zedelgem, Belgium). Man–Rogosa–Sharpe (MRS) culture medium and sodium standards (1000 g/L) were acquired from PanReac AppliChem (Castellar del Valles, Spain). Ultrapure Milli-Q (Millipore, Darmstadt, Germany) Type 1 water (<0.054 μS/cm) was used in all experiments.

### 3.2. Centesimal Composition and Pigments

Centesimal composition of leaves, flowers, and schizocarps of *C. maritimum* was determined as described by Primitivo et al. [34] with few modifications. Briefly, moisture content was obtained by weighting 5.0 g of fresh samples into porcelain crucibles, followed by oven dehydration at 105 °C (Heraeus D-6450, Singapore) until constant weight. Moisture content was calculated by gravimetry and expressed as a percentage of fresh weight (% fw). For ash determination, dehydrated samples were incinerated in a muffle (Nabertherm, Liliemthal/Bremen, Germany), at 500 °C, for 12 h. Ash content was calculated as a percentage of the initial dry weight (% dw).

Total protein content was determined by quantifying the nitrogen content of the samples using the Kjeldahl method [49], with a conversion factor of 6.25. Freeze-dried samples (0.5 g) were digested with 70 mL of sulfuric acid (97% *v*/*v*) and a catalyst tablet for 90 min at 400 °C. Cooled samples were mixed with 70 mL of deionized water and subjected to alkaline distillation using a Kjeltec 2100 apparatus (Foss, Denmark). Distillates were collected in boric acid (30 mL at 4.0% *v*/*v*) and titrated with standard 0.1 M hydrochloric acid (HCl). Total protein content was calculated as described by Primitivo et al. [34] and expressed as a percentage of dry weight (% dw).

Lipid content was determined using the Folch gravimetric method [49] with modifications as described elsewhere [34]. Briefly, freeze-dried samples (1 g) were mixed with 10 mL of a solvent mixture containing chloroform (CHCl_3_) and methanol (CH_3_OH) in a 2:1 ratio and stirred for 5 min. Next, 1.2 mL of 0.8% (*w*/*v*) NaCl was added, and the mixture was vortexed and subjected to centrifugation (at 4000× *g* for 5 min) to facilitate phase separation. Lower phases were collected and filtered through an anhydrous sodium sulfate column into a pre-weighed round-bottom flask. Subsequently, 5 mL of chloroform was added to the remaining mixtures, and the extraction procedure was repeated. The solvent was then removed by evaporation using a rotary evaporator (Heidolph 2, LAB1ST, Shanghai, China), and the lipidic residues were dried at 40 °C until constant weight. Total fat content was expressed as % dw. Total carbohydrates (% dw) were calculated by difference, subtracting the sum of the percentages of total protein, fat, and ash from 100% according to the AOAC [49].

Chlorophylls and carotenoids were quantified as described by Brito et al. [36]. Briefly, freeze-dried samples (0.25–0.5 g) were macerated with pure acetone (≥99.5% *v*/*v*), and volumes adjusted to 10.0 mL. Then, absorbances of the extracted solutions were measured at 470, 644.8, and 661.6 nm (Thermo Scientific, Evolution 201 UV–VIS Spectrophotometer), and the concentration of chlorophylls and carotenoids were calculated as described by Lichtenthaler and Buschmann [50].

### 3.3. Profile of Minerals

Mineral contents were determined by micro-energy dispersive X-ray fluorescence (µ-EDXRF) (Bruker M4 Tornado^TM^, Leipzig, Germany) equipped with a low-power HV generator and a Rh-anode X-ray tube. Dehydrated samples were macerated and transformed into pellets of 2 cm in diameter and 1 mm in thickness. An XFlash^®^ SDD detector was used for detection with energy resolutions above 145 eV and spectral acquisition times of 1000 s. Elemental maps were used to produce average spectra. Spectra quantification was performed with the built-in ESPRIT software v1.0. Recovery rates were checked against standard samples: bush branches, orchard leaves, and poplar leaves with the codes GBW 07603, NBS 1571, and GBW 07604, correspondingly. Detection limits of Rb, Sr, K, Ca, Mn, and Fe were 0.8, 1.0, 55, 35, 4.0, and 15 µg/g, respectively. The detection limit of Cu, Ni, and Zn was 2.0 µg/g. Results were expressed as µg/g dw. Given the high amount of sodium (Na) present in the plant material, contents in Na were obtained by atomic absorption spectrometry (Varian-SpectrAA-55B, Grenoble, France) with an air and acetylene oxidizing flame. For the AAS analysis, freeze-dried plant material (120–130 mg) was digested with 10 mL of nitric acid and 5 mL of sulfuric acid. Calibration was performed with a mono elemental lamp at 589.6 nm 0–5.0 mg/L with a detection limit of 0.05 mg/L. Results were expressed as µg/g dw.

### 3.4. Profile of Fatty Acids

Fatty acid profiles of leaves, flowers, and schizocarps of *C. maritimum* were analyzed by gas chromatography, as described by Brito et al. [36]. Briefly, fatty acid methyl esters (FAME) were obtained by direct acid transmethylation (2.0% *v*/*v* of H_2_SO_4_ in methanol, at 80 °C for 2 h) of freeze-dried samples and collected in *n*-hexane. C17 FAME was used as an internal standard. Analyses were performed in a Finnigan Ultra Trace gas chromatograph, equipped with a Thermo TR-FAME capillary column 60 m in length, 0.25 mm in diameter, and 0.25 µm in membrane thickness, an AS 3000 auto sampler (Thermo Electron Corporation, Waltham, MA, USA) and a flame ionization detector (FID). The chromatographic conditions of the FID detector (280 °C, supplied with synthetic air and hydrogen), injector (250 °C, in splitless mode), and oven temperature program were as previously described by Primitivo et al. [34]. Xcalibur software Version 1.4 (Thermo Fisher Scientific Inc., USA) was used for data acquisition and analysis. Fatty acids (FAs) were identified by comparison of retention times with standard samples (Supelco 37, PUFA1, and PUFA3). Results were expressed as a percentage of total area (% Total FAs) and as absolute concentration (µg/mg dw), calculated by the internal standard method, as described previously [34]. Lipid quality parameters: hypocholesterolemic and hypercholesterolemic FAs ratio (h/H), atherogenicity index (AI), and thrombogenicity index (TI) were calculated, as described by Ulbricht and Southgate [51].

### 3.5. Total Phenolic Content and Antioxidant Capacity

Plant extracts were produced and tested for phenolic compounds by the Folin–Ciocalteu assay and for antioxidant capacity by three different assays, based on the reduction of the ABTS●+ and DPPH● radicals and on the reduction of ferric (III) to ferrous iron (II) (FRAP, ferric-reducing antioxidant power). Plant extracts of leaves, flowers, and schizocarps of *C. maritimum* were obtained using a sequential extraction method with solvents of increasing polarity, as previously performed for other halophytes [52]. First, dried plant material was extracted with petroleum ether, followed by acetone, ethanol, a 1:1 (*v*/*v*) mixture of ethanol and water, and finally, water. The solid-to-liquid ratio was 1:10 g/mL, and the extraction took place under agitation at 9000 rpm, at 20 °C, for 4 h. The resulting extracts were subjected to evaporation using a rotary evaporator (Heidolph 2, LAB1ST, Shanghai, China) in the case of solvents being petroleum ether, acetone, and ethanol. For solvents containing water, extracts were first evaporated and then lyophilized. This methodology allowed us to obtain five extracts of each part of the plant with varying degrees of polarity, which were subsequently analyzed in terms of total phenolic content and antioxidant capacity. For these analyses, extract solutions at 500 or 1000 µg/mL, prepared with the respective extraction solvent, were used.

Total phenolic compounds were determined by the micro-scale modified Folin–Ciocalteu colorimetric method, as described previously [53], with few modifications. Briefly, in Eppendorf tubes, aliquots (10 µL) of the extracts were added to 840 µL of Milli-Q water, 50 µL of Folin–Ciocalteu reagent, and 150 μL of 20% (*w*/*v*) Na_2_CO_3_. After vortex stirring, mixtures were kept in the dark at room temperature for 1 h, and absorbances were measured at 755 nm in quadruplicate using a microplate reader (Biotek, Epoch2C, Agilent, Santa Clara, CA, USA). Solutions of gallic acid (GA) ranging from 0 to 175 µg/mL underwent the same process and were used to build a standard curve. Results were expressed as micrograms of GA equivalents per milligram of extract (µg GAE/mg).

The ABTS scavenging activity was evaluated as described by Antunes et al. [54]. For the color reaction, 50 µL of the extracts were mixed with 950 µL of ABTS reagent (7.0 mM (NH_4_)_2_C_18_H_16_N_4_O_6_S_4_ and 12.25 mM K_2_S_2_O_8_ in a ratio of 4:1). Absorbance of the resulting solutions was measured at 734 nm (Varian Cary 50 UV–Visible spectrophotometer, Varian, Grenoble, France). The same procedure was repeated for the baseline solutions (prepared with the extraction solvents) and for the standard solutions used to build a calibration curve (Trolox concentrations ranging from 80 to 10 µg/mL). Results were expressed as micrograms of Trolox equivalents per milligram of extract (µg Trolox/mg).

For the DPPH radical scavenging activity analysis, extracts (50 µL) were added to 950 µL of DPPH (40 µg/mL in ethanol). After vortex stirring, the mixtures were kept in the dark for 30 min, and their absorbances (517 nm) were read in quadruplicate in a microplate reader (Biotek, Epoch2C). The maximum absorbance was attained with mixtures of 950 µL of DPPH with 50 µL of each extraction solvent. A calibration curve was prepared with Trolox solutions (from 50 to 250 µg/mL), and results were expressed as Trolox equivalents (μg Trolox/mg) as reported by Antunes et al. [54].

The FRAP assay was conducted according to Neves et al. [53] with few modifications. Extract solutions (30 μL) were mixed with 900 μL of FRAP reagent (0.3 mol/L acetate buffer, 20 mmol/L FeCl_3_, and 10 mmol/L TPTZ in a ratio of 10:1:1). After 40 min in the dark for color development, 200 μL aliquots were transferred to a 96-well microplate, and absorbance was measured at 595 nm. Calibration curves were prepared using standard solutions of Trolox (ranging from 1.7 to 170 μg/mL). Antioxidant activity was expressed as Trolox equivalents (μg Trolox/mg).

### 3.6. NMR Experiments

One-dimensional (1D) and two-dimensional (2D) nuclear magnetic resonance experiments (NMR) were conducted on a Bruker Avance III 400 spectrometer (Bruker, MA, USA) operating at a ^1^H frequency of 400.133 MHz. Samples (0.5 g) of plant material were freeze-dried (Kinetics Ez-Dry, Desden, Germany). NMR solutions were prepared by dissolving freeze-dried material in deuterated solvents and centrifuging at 4000× *g* (Heraeus Labofuge 200, Waltham, MA, USA) for 5 min. One-dimensional ^1^H NMR spectra were acquired with a spectral width of 4401.41 Hz, and *zgpr* pulse sequences were collected with 32 k complex points and 64 scans, using a recycle delay of 2 s at 25 °C. Two-dimensional [^1^H-^1^H]-COSY (correlation spectroscopy) NMR experiments used gradient pulses for selection and presaturation during relaxation for solvent signal suppression. Two-dimensional [^1^H-^1^H]-COSY spectra were acquired with 2 k complex points in t2 and 512 increments (16 scans), a spectral width of 4401.41 Hz (11 ppm) in both dimensions, and a relaxation delay of 1.4 s. SINE window functions were used for spectra apodization. Zero filling to 1 k points was applied in both dimensions. Bruker Topspin v4.0 and MestReNova 9.1 software were used for data processing and analysis. Chemical shifts of external standards were used for referencing (in ppm).

### 3.7. Antimicrobial Activity

#### 3.7.1. Antibacterial Activity

Two-fold broth microdilution assays were performed to determine the antibacterial activity of the plant extracts against *E. coli* ATCC 25922, *K. pneumoniae* ATCC 27799, *S. aureus* ATCC 29213 and *S. epidermidis* ATCC 12228, according to Mogana et al. [55]. Plant extracts were prepared by using dissolved extracts in DMSO in the case of extractions with petroleum ether, acetone, or in water when solvents were 1:1 ethanol/water and water to a final concentration of 3000 µg/mL. Concentrations of plant extracts ranged from 2.34 to 600 µg/mL in Mueller–Hinton broth (MHB). Each microtiter plate was inoculated with a standardized inoculum of each microorganism, prepared as described previously [55]. Plates were incubated at 37 °C for 24 h, and microbial growth was assessed by visual inspection. To ensure the validity of the results, sterility controls (MHB without inoculation) and growth controls (MHB with inoculation) were included for each strain in all plates. In addition, quality controls were performed using *E. coli* ATCC 25922 and *S. aureus* ATCC 29213 to determine the MIC of cefotaxime against each strain, according to EUCAST (eucast.org, accessed on 15 June 2022). Extract concentrations with no microbial growth were used as the minimal inhibitory concentration (MIC). Minimal bactericidal concentrations (MBC) were determined by plating 10 µL of broth from wells with no visible growth on the MH agar. After incubating for 24 h, the smallest concentration that presented no growth was considered to be the MBC.

#### 3.7.2. Antifungal Activity

Antifungal activity of *C. maritimum* extracts was evaluated against *C. albicans* ATCC 10259, *C. parapsilosis* ATCC 22019, *S. cerevisiae* ATCC 18824, and a wild strain of *K. phaffii* using a two-fold broth microdilution assay, following the methodology described by Rodríguez-Tudela et al. [56]. The antifungal activity was determined in RPMI 1640 synthetic medium, without bicarbonate, supplemented with glutamine, and with a final concentration of 2% (*w*/*v*) of glucose. MIC determinations of *C. albicans* and *C. parapsilosis* yeasts were obtained at 37 °C, while *S. cerevisiae* and *K. phaffii* were conducted at 28 °C for 24 h. Concentrations of plant extracts were the same as those used for the determination of antibacterial activity. Sterility and growth controls were performed as described in the above section, and quality controls were conducted by determining the MIC of *C. albicans* ATCC 10259 to fluconazole and considering the epidemiological cut-off values (ECOFF) according to EUCAST guidelines. Minimal fungal concentrations were determined as described for the determination of MBC, using a universal medium for *S. cerevisiae* ATCC 18824, *K. phaffii*, and *C. parapsilosis* ATCC 22019, while potato dextrose agar was used for *C. albicans* ATCC 10259. Incubations took place at the same temperatures used for MIC determination.

### 3.8. MTT Cell Viability Tests

Caco-2 intestinal epithelial model cells were exposed to plant material. MTT cell viability tests were used to assess cell viability. Cells were maintained in Dulbecco’s modified Eagle’s medium (DMEM) supplemented with 10% (*v*/*v*) heat-inactivated fetal bovine serum (FBS), 1% (*v*/*v*) non-essential amino acids, and 1% penicillin–streptomycin, in a humidified atmosphere of 5% CO_2_, at 37 °C. Cells were subcultured twice a week at 80–90% confluence by making use of cell detachment of a 2.5% (*w*/*v*) trypsin-EDTA solution (10×). Cells were then seeded in 96-well microplates at a density of 1 × 10^4^ cells per well, andcells were maintained in culture for 4 days. DMEM medium renewals were performed every two days. To assess toxicity via cell viability, cells were incubated for 24 h with increasing concentrations of freeze-dried macerated sea fennel leaves, previously added to DMEM medium at concentrations ranging from 0.5 to 5.0% (*w*/*v*). After exposure, cells were washed with PBS buffer and incubated for 4 h with 0.5 mg/mL MTT at 37 °C. Upon incubation, MTT solutions were discarded. The formazan precipitates formed were dissolved in DMSO. Quantification was performed through UV–visible spectroscopy by measuring the absorbance of the resulting solutions at 570 nm (Microplate reader BioTek Synergy HT, Biotek Instruments, Winooski, VT, USA). Cells incubated with a DMEM medium were uniquely used to determine the absorbance related to 100% cell viability.

### 3.9. Prebiotic Effect on Lactobacillus bulgaricus

The potential prebiotic effect of sea fennel leaves was investigated on the growth of the probiotic *Lactobacillus bulgaricus*. Commercial *Lactobacilli* (Naturitas, Cabestany, France) were selectively grown in MRS broth agar plates at 37 °C for 24 h under aerobic conditions. Monocultures were transferred and grown in MRS broth liquid media at 37 °C for 12 h. Bacterial growth was monitored on 1:500 diluted inocula at 37 °C for 12 h in the absence and presence of freeze-dried macerated *C. maritimum* leaves at 0.1% (*w*/*v*). Culture supernatants were used for optical density measurements at 600 nm.

### 3.10. Statistical Analysis

All experiments were performed at least in triplicate. Results are reported as mean ± standard deviation (SD). Data were analyzed either by one-way or two-way analysis of variance (ANOVA). Effects were deemed statistically significant at *p* < 0.05. Correlations were based on Pearson’s coefficients. Calculations were performed with the *R* software (Version 4.3.2 Vienna, Austria).

## 4. Conclusions

Edible leaves, flowers, and herbs can act as natural food additives and preservatives, replacing other regulatory-approved synthetic chemicals that are detrimental to human/animal health. The edible halophyte *C. maritimum* (sea fennel) is a source of minerals, omega-6, and omega-3 PUFAs, polyphenols, and other bioactive compounds with relevant antibacterial, antifungal, and antioxidant properties.

*C. maritimum* aerial parts (flowers, leaves, and fruits/schizocarps) present high moisture (>85%) and are mainly composed of carbohydrates (>60%), followed by ash (~20%), protein (~7%), and crude fat (~5%). Moreover, while sea fennel flowers are richer in proteins, leaves are richer in ash, and schizocarps are richer in lipids. Sea fennel aerial parts are also particularly rich in macrominerals such as Na, Ca, Cl, K, P, and S. Biological pigments (chlorophylls, xanthophylls, and carotenes) are present in all sea fennel’s aerial parts being chlorophyll a particularly abundant in the leaves. Regarding lipids, all plant parts are mainly composed of unsaturated fatty acids (>65% of total fatty acids). While leaves and flowers are mainly composed of polyunsaturated fatty acids (PUFAs) (>50% of total fatty acids), schizocarps contain mostly monounsaturated (MUFAs) fatty acids (>60% of total fatty acids).

In addition, ethanolic extracts of sea fennel flowers present strong antioxidant activity attributed to the presence of phenolic acids, flavanols, and anthocyanins. On the other hand, apolar extracts of *C. maritimum* flowers and schizocarps present antimicrobial activity against Gram-positive bacteria (*Staphylococcus aureus* and *Staphylococcus epidermidis*) and against fungi (*Candida albicans*, *Candida parapsilosis*, *Komagataella phaffii* and *Saccharomyces cerevisiae*), with low toxicity to human intestinal epithelial Caco-2 model cells for <5.0% *w*/*v* plant material.

Hence, the rational use of sea fennel in the agri-food sector can bring nutrients, aroma, and flavor to culinary dishes while contributing to regulating microbiota and expanding the shelf life and safety of food products.

## Figures and Tables

**Figure 1 plants-13-00427-f001:**
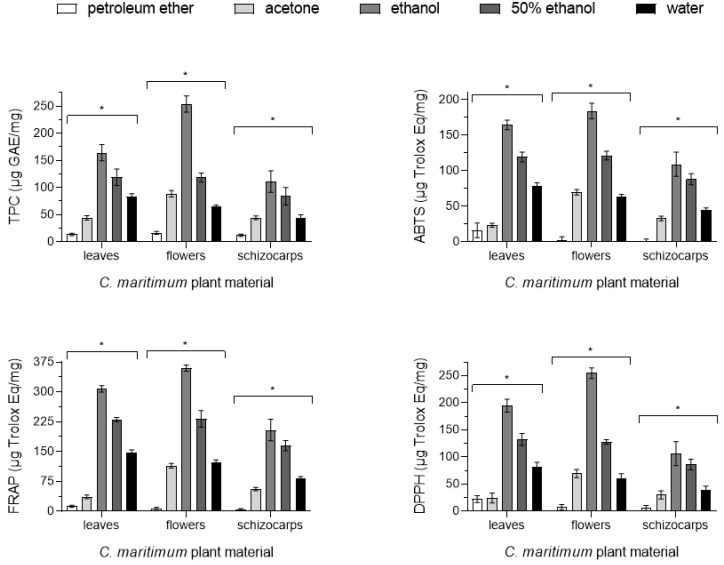
Antioxidant capacity and total phenolic content of *C. maritimum*. Total phenolic content (TPC) expressed as gallic acid equivalents (μg GAE/mg), ABTS radical scavenging activity, ferric-reducing activity power (FRAP), and DPPH scavenging activity expressed as Trolox equivalents (Trolox Eq μg/mg) of the extracts sequentially obtained (in petroleum ether, followed by acetone, ethanol, ethanol/water, and water) of leaves, flowers, and fruits/schizocarps of sea fennel. Results are expressed as mean ± standard deviation. A two-way ANOVA was applied to analyze differences between solvents and biochemical methods for the different types of plant material. * Significant differences (*p* < 0.05).

**Figure 2 plants-13-00427-f002:**
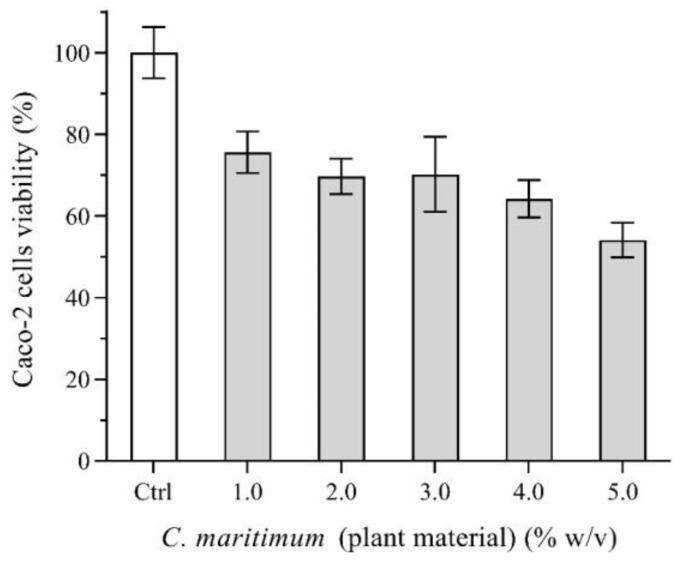
MTT viability assay of Caco-2 cells. Cells were exposed to increasing concentrations of *C. maritimum* (sea fennel) plant material in DMEM culture medium for 24 h. Control (Ctrl) refers to DMEM medium and 100% viability. Results are expressed as mean ± standard deviation.

**Table 1 plants-13-00427-t001:** Centesimal composition and contents in pigments of sea fennel leaves, flowers, and fruits/schizocarps, and mineral composition of whole aerial parts in Ca, Cl, Cu, Fe, K, Mn, Na, P, S, and Zn (µg/g dw).

*Crithmum maritimum* L.—Sea Fennel					Leaves		Flowers	Schizocarps	
Moisture (%)					88.4 ± 0.7 ^a^		85.0 ± 0.1 ^b^	86.8 ± 0.3 ^c^	
Centesimal composition (% dw)		Ash			24.9 ± 0.8 ^a^		15.9 ± 0.3 ^b^	19.6 ± 0.5 ^c^	
		Crude fat			4.0 ± 0.1 ^a^		4.2 ± 0.1 ^a^	6.5 ± 0.5 ^b^	
		Proteins			4.6 ± 0.1 ^a^		8.3 ± 0.2 ^b^	7.1 ± 0.3 ^c^	
		Carbohydrates			66.5 ± 0.7 ^a^		71.7 ± 0.4 ^b^	66.7 ± 0.4 ^a^	
Pigments (µg/g dw)		Chlorophyll *a*			855.8 ± 6.0 ^a^		197.5 ± 4.6 ^b^	132.0 ± 1.6 ^c^	
		Chlorophyll *b*			236.5 ± 2.1 ^a^		56.8 ± 2.4 ^b^	43.1 ± 1.0 ^c^	
		Xanthophylls and Carotenes			258.0 ± 2.0 ^a^		86.1 ± 1.1 ^b^	63.6 ± 0.7 ^c^	
Ca	Cl	Cu	Fe	K	Mn	Na	P	S	Zn
1180 ± 10	2980 ± 20	1.30 ± 0.24	43.9 ± 1.72	1600 ± 10	16.7 ± 1.24	36,500 ± 5000	335 ± 49.0	983 ± 55.2	5.02 ± 0.42

Values expressed as mean ± standard deviation (*n* = 3). Different letters across the columns mean significant differences (*p* < 0.05).

**Table 2 plants-13-00427-t002:** Total phenolic content (TPC) expressed as gallic acid equivalents (μg GAE/mg), ABTS radical scavenging activity, ferric-reducing activity power (FRAP), and DPPH scavenging activity expressed as Trolox equivalents (Trolox Eq μg/mg) of the several extracts (petroleum ether, acetone, ethanol, aqueous-ethanolic, and aqueous) of leaves, flowers, and fruits/schizocarps of sea fennel, *Crithmum maritimum* L.

Extract	Petroleum Ether	Acetone	Ethanol	50% Ethanol	Water
TPC (µg GAE/mg)					
Leaves	13.6 ± 2.3 ^a^	44.0 ± 4.3 ^b^	164 ± 15 ^c^	119 ± 15 ^d^	83.2 ± 5.3 ^e^
Flowers	16.2 ± 2.6 ^f^	88.4 ± 6.1 ^g^	254 ± 15 ^h^	118.4 ± 8.2 ^i^	64.2 ± 3.8 ^j^
Schizocarps	11.7 ± 2.3 ^k^	44.2 ± 3.4 ^l^	111 ± 20 ^m^	84 ± 16 ^n^	44.3 ± 5.3 ^o^
ABTS (µg Trolox Eq/mg)					
Leaves	16.0 ± 10.3 ^a^	23.2 ± 2.9 ^b^	164.4 ± 6.6 ^c^	119.2 ± 7.0 ^d^	78.6 ± 4.4 ^e^
Flowers	2.5 ± 4.6 ^f^	69.7 ± 3.9 ^g^	184 ± 11 ^h^	121.2 ± 6.1 ^i^	62.7 ± 3.8 ^j^
Schizocarps	1.0 ± 2.7 ^k^	32.8 ± 3.2 ^l^	109 ± 17 ^m^	87.8 ± 7.8 ^n^	44.7 ± 2.8 ^o^
FRAP (µg Trolox Eq/mg)					
Leaves	12.6 ± 2.8 ^a^	36.2 ± 4.9 ^b^	307.4 ± 8.2 ^c^	230.8 ± 5.2 ^d^	147.2 ± 7.4 ^e^
Flowers	7.4 ± 2.7 ^f^	114.3 ± 6.1 ^g^	359.8 ± 7.7 ^h^	232 ± 21 ^i^	121.7 ± 7.6 ^j^
Schizocarps	5.2 ± 1.6 ^k^	55.9 ± 4.3 ^l^	204 ± 27 ^m^	165 ± 13 ^n^	83.1 ± 4.0 ^o^
DPPH (µg Trolox Eq/mg)					
Leaves	22.1 ± 6.4 ^a^	24.4 ± 9.1 ^b^	194.6 ± 12.0 ^c^	132.2 ± 10.8 ^d^	80.9 ± 9.3 ^e^
Flowers	7.5 ± 4.6 ^f^	69.3 ± 7.5 ^g^	254.5 ± 9.9 ^h^	127.7 ± 4.2 ^i^	59.8 ± 9.2 ^j^
Schizocarps	5.9 ± 4.7 ^k^	29.9 ± 7.5 ^l^	106 ± 22 ^m^	86 ± 10 ^n^	38.4 ± 7.8 ^o^

Values in mean ± SD of triplicates of independent samples. Different letters across the lines and columns mean significant differences (*p* < 0.05). GAE, gallic acid equivalents.

**Table 3 plants-13-00427-t003:** Fatty acid profile of leaves, flowers, and fruits/schizocarps of sea fennel, *C. maritimum*.

	Leaves		Flowers		Schizocarps	
	% Total FA	µg/mg dw	% Total FA	µg/mg dw	% Total FA	µg/mg dw
SFA						
C12:0	0.45 ± 0.29	0.077 ± 0.064	0.066 ± 0.03	0.011 ± 0.005	0.02 ± 0.01	0.013 ± 0.008
C14:0	2.49 ± 0.09	0.429 ± 0.019	0.547 ± 0.003	0.142 ± 0.008	0.27 ± 0.01	0.131 ± 0.042
C15:0	0.22 ± 0.06	0.030 ± 0.004	0.18 ± 0.01	0.048 ± 0.005	0.11 ± 0.00	0.055 ± 0.019
C16:0	18.53 ± 0.18	3.210 ± 0.130	17.57 ± 0.10	4.570 ± 0.270	8.89 ± 0.12	4.359 ± 1.440
C17:0	0.25 ± 0.01	0.043 ± 0.001	0.225 ± 0.003	0.059 ± 0.004	0.12 ± 0.00	0.058 ± 0.020
C18:0	7.97 ± 0.08	1.377 ± 0.062	3.54 ± 0.02	0.918 ± 0.051	0.42 ± 0.03	0.200 ± 0.080
C20:0	1.43 ± 0.02	0.246 ± 0.011	0.72 ± 0.01	0.186 ± 0.014	0.28 ± 0.01	0.134 ± 0.044
C24:0	2.12 ± 0.07	0.364 ± 0.006	2.64 ± 0.02	0.684 ± 0.038	1.00 ± 0.02	0.49 ± 0.16
MUFA						
C15:1	0.32 ± 0.05	0.059 ± 0.006	0.17 ± 0.02	0.043 ± 0.008	0.07 ± 0.01	0.035 ± 0.016
C16:1	1.43 ± 0.26	0.220 ± 0.018	0.27 ± 0.01	0.067 ± 0.004	0.16 ± 0.01	0.076 ± 0.026
C17:1	0.22 ± 0.01	0.039 ± 0.002	0.53 ± 0.01	0.138 ± 0.010	0.07 ± 0.02	0.040 ± 0.012
C18:1 n-9	2.03 ± 0.07	0.355 ± 0.013	3.43 ± 0.03	0.889 ± 0.043	50.1 ± 1.00	29.7 ± 2.00
C18:1 n-7	0.44 ± 0.05	0.072 ± 0.008	0.52 ± 0.01	0.134 ± 0.007	0.48 ± 0.06	0.24 ± 0.06
C22:1 n-9	6.62 ± 0.07	1.139 ± 0.052	10.04 ± 0.17	2.610 ± 0.160	7.05 ± 0.30	3.43 ± 1.05
C24:1 n-9	2.63 ± 0.07	0.456 ± 0.015	4.12 ± 0.06	1.074 ± 0.076	3.07 ± 0.03	1.51 ± 0.51
PUFA						
C16:3 n-4	0.33 ± 0.01	0.057 ± 0.002	0.07 ± 0.01	0.015 ± 0.003	0.07 ± 0.00	0.033 ± 0.009
C18:2 n-6	29.47 ± 0.39	5.11 ± 0.33	45.25 ± 0.24	11.77 ± 0.64	23.70 ± 0.50	11.6 ± 3.8
C18:3 n-3 (ALA)	22.50 ± 0.36	3.90 ± 0.24	8.54 ± 0.04	2.22 ± 0.12	2.67 ± 0.10	1.30 ± 0.41
C20:5 n-3 (EPA)	0.56 ± 0.04	0.091 ± 0.001	1.58 ± 0.02	0.406 ± 0.025	1.51 ± 0.02	0.74 ± 0.25
SFA	33.45 ± 0.44		25.48 ± 0.11		11.10 ± 0.14	
MUFA	13.69 ± 0.30		19.08 ± 0.20		60.95 ± 0.76	
PUFA	52.86 ± 0.63		55.44 ± 0.27		27.95 ± 0.62	
n-3	23.07 ± 0.35		10.12 ± 0.04		4.18 ± 0.12	
n-6	29.47 ± 0.39		45.25 ± 0.24		23.70 ± 0.50	
n-3/n-6	0.78 ± 0.01		0.22 ± 0.00		0.176 ± 0.001	
H/H	2.57 ± 0.06		3.16 ± 0.03		8.35 ± 0.16	
AI	1.247 ± 0.020		0.992 ± 0.007		0.405 ± 0.006	
TI	0.317 ± 0.005		0.345 ± 0.002		0.174 ± 0.002	

Values in mean ± SD of triplicates of three independent samples. FA, fatty acid. SFA, saturated fatty acid. MUFA, monounsaturated fatty acid. PUFA, polyunsaturated fatty acid. dw, dry weight.

**Table 4 plants-13-00427-t004:** Antimicrobial MIC, MBC, and MFC of extracts of leaves, flowers, and fruits/schizocarps of sea fennel, *Crithmum maritimum* L. in different solvents (petroleum ether, acetone, and ethanol).

		**MIC (µg/mL)**					
	**Leaves**	**Flowers**			**Schizocarps**		
Bacteria	acetone	petrol. ether	acetone	ethanol	petrol. ether	acetone	ethanol
*E. coli*	>600	>600	>600	>600	>600	>600	>600
*K. pneumoniae*	>600	>600	>600	>600	>600	>600	>600
*S. aureus*	>600	75	75	300	75	75	300
*S. epidermidis*	>600	150	>600	300	150	>600	600
Fungi							
*C. albicans*	300	150	75	>600	75	75	600
*C. parapsilosis*	>600	300	300	>600	150	300	>600
*K. phaffii*	>600	150	>600	>600	300	150	>600
*S. cerevisiae*	>600	>600	>600	>600	300	300	>600
		**MBC (µg/mL)**					
	**Leaves**	**Flowers**			**Schizocarps**		
Bacteria	acetone	petrol. ether	acetone	ethanol	petrol. ether	acetone	ethanol
*E. coli*	>600	>600	>600	>600	>600	>600	>600
*K. pneumoniae*	>600	>600	>600	>600	>600	>600	>600
*S. aureus*	300	300	150	>600	150	150	>600
*S. epidermidis*	600	300	300	>600	300	300	>600
**MFC (µg/mL)**							
Fungi							
*C. albicans*	>600	300	150	>600	150	150	>600
*C. parapsilosis*	>600	300	300	>600	300	300	>600
*K. phaffii*	600	150	150	>600	150	150	600
*S. cerevisiae*	>600	600	>600	>600	600	300	>600

MIC, minimal inhibitory concentration; MBC, minimal bactericidal concentration; MFC, minimal fungicidal concentration. Extracts were tested against all strains as outlined in the Section 3. Extracts not listed in the table correspond to cases where MIC, MBC, or MFC values were >600 µg/mL.

## Data Availability

The data presented in this study are available in the article and Appendix A here.

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
