# Peer review of "Nutritive Value and Bioactivities of a Halophyte Edible Plant: *Crithmum maritimum* L. (Sea Fennel)"

_plants, 2024, doi:10.3390/plants13030427_

Round 1
Reviewer 1 Report
Comments and Suggestions for Authors
Materials and Methods - are well described
I recommend an improvement of the conclusions
Author Response
Thank you for your review and comments.
The final section (Conclusions) is now more elaborated and comprehensive (4 paragraphs - 300 words).
Reviewer 2 Report
Comments and Suggestions for Authors
The manuscript is very interesting and edited correctly. In my opinion, the research results obtained are valuable. I recommend the paper for publication in Plants in its present form.
Author Response
Thank you for the detailed analysis and positive comments.
Reviewer 3 Report
Comments and Suggestions for Authors
A complete and thorough characterization of the Crithmum maritimum L. (sea-fennel) extracts.
The notes where you specify thet "different letters mean..." should include "across the columns" or "across the lines".
In 3.10 Statistical analysis, you mentioned the use of two-way ANOVA statistical method. Please define the two factors and their levels. Also, you have to include, where was used, in the table captions the use of 2-way ANOVA.
For Figure 1, you should have statistical analysis like 1-way ANOVA results. Because of the great work you done, the conclusion should be complete with those results.
I know that the journals politics is that results should be displayed as numeric (Tables) or graphical (Figures). Because your work involve huge number of variables and sample groups, I suggest to append in the supplementary document the graphics that emerge from the tables. In this way the readers will follow easily the discussions and the conclusions. This is an optional issue to aleviate, but the readers will understand faster.
Author Response
Thank you for the detailed analysis, suggestions, and positive comments.
Yes, regarding the statistical analyses, we have introduced the expression “across the columns/lines” in figure legends and table legends, as well as the factors used in the 1-way and 2-way ANOVAs.
Yes, to better depict and present the results, we have included two more sets of graphs in the paper (Fig.1- main manuscript and Fig. S1 – Supp. Material). Figure 1 (with 4 panels A to D) presents the total phenolic contents (TPC – Folin–Ciocalteu) and antioxidant capacity (AC by the FRAP, ABTS and DPPH assays) determined for the different extracts obtained from the different plant parts (leaves, flowers and fruits), in the different solvents (with increasing polarity). Figure S1 (with 3 panels) presents the centesimal composition, mineral profile and content in pigments detected in C. maritimum leaves, flowers, and fruits/schizocarps.
Reviewer 4 Report
Comments and Suggestions for Authors
Minor remarks
1. All minor remarks are depicted in the manuscript file.
2. Please, provide a blank space between quantity and unit except in the case of percentage.
3. Use mL for milliliters.
4. The English language should be improved. Some minor grammatical errors need further revision process.
5. Avoid the use of the first-person plural. Only the third-person singular is acceptable for the scientific paper.
6. Latin terms and Greek symbols should be presented in italics. Please, carefully check it in the references list also.
7. Please, keep in mind that the already defined abbreviations should be used in the manuscript. Also, I have noticed that some abbreviations are presented almost at the end of the manuscript, in the Experimental. The terms should be defined after the first mention in the paper.
Major remarks
1. The main weakness of this manuscript is the lack of interesting figures. The results should be depicted in an interesting way to attract the attention of readers. Consider inserting some schematical representation of carried out experiments, i.e., to summarize all done.
2. The conclusion section is very poor in its present form. It should provide a concise summary of the key findings, their significance, their implications, and a sense of closure to the study. Please, retype this section to contain all important data concluded from this study.

English language is well-written but some minor grammatical mistakes need to be corrected.
Author Response
Thank you for the detailed analysis, suggestions, and comments.
[Minor remarks: 1 to 7] The manuscript has been revised accordingly. All scientific names and Latin expressions are now in italic. All units are correct.
[Major remark 1] - Yes, to better depict and present the results, we have included two more sets of graphs in the paper (Fig.1- main manuscript and Fig. S1 – Supp. Material). Figure 1 (with 4 panels A to D) presents the total phenolic content (TPC – Folin–Ciocalteu) and antioxidant capacity (AC by the FRAP, ABTS and DPPH assays) determined for the different extracts obtained from the different plant parts (leaves, flowers, and fruits/schizocarps), in the different solvents (with increasing polarity). Figure S1 (with 3 panels) presents the centesimal composition, mineral profile and content in pigments detected in C. maritimum leaves, flowers, and fruits/schizocarps. We have also included a graphical abstract summarizing all analyses performed and main results obtained.
[Major remark 2] - Yes, the final section has been updated. The “Conclusions” section is now more elaborated and comprehensive (4 paragraphs - 300 words), as it summarizes the key findings and their implications.